# Four Centuries of Medicinal Mosses and Liverworts in European Ethnopharmacy and Scientific Pharmacy: A Review

**DOI:** 10.3390/plants10071296

**Published:** 2021-06-25

**Authors:** Jacek Drobnik, Adam Stebel

**Affiliations:** Department of Pharmaceutical Botany, Faculty of Pharmaceutical Sciences in Sosnowiec, Medical University of Silesia in Katowice, Ostrogórska 30, 41–200 Sosnowiec, Poland; astebel@sum.edu.pl

**Keywords:** bioprospecting of medicinal plants, historical remedies, mosses, liverworts, hornworts

## Abstract

(1) Medicinal use of bryophytes dates to ancient times, but it has always been marginal due to their small size, difficult identification, lack of conspicuous organs which would attract attention (flowers, fruits) and insipid taste of the herb. The earliest testimonies of their medical use come from the 1500s. The interest in medicinal bryophytes diminished considerably in the 1880s, except for *Sphagnum* spp., which became a source of dressing material. The second half of the 20th century saw the revival of the study of bryophyte chemistry. (2) Historical printed sources from 1616 to 1889 were queried. Bryophyte species found were taxonomically identified and presented against the background of their confirmed properties and ecology. The study was supplemented with historical vs. modern ethnomedicinal data. (3) In 26 publications, 28 species were identified. Modern usage was known for 10 of them. Medicinal properties of 16 species were confirmed. (4) Species of wide geographical distribution range were (or are still being) used in local folk medicines. Historical ethnobiological and ethnopharmaceutical uses of them are sometimes convergent with their confirmed properties, mostly external (as antimicrobial or cytotoxic remedies).

## 1. Introduction

Bryophytes, as any other plant, drew the interest of modern-period botanists in the early 16th century. European herbaria created between ca. 1542–1577 contain as many as 34 identifiable species of bryophytes, collected intentionally (i.e., excluding “by-catches”), and representatives of eight further genera [1]. In the same period, bryophytes must have been also considered as potential medicinal plants. Fuchs in 1549 [2] (p. 600) was able to propose a new (i.e., unknown to Dioscorides) moss species, *Polytrichum*, as a candidate for a medicinal species [3] and described its usage this way: “A decoction of this plant in water or lye strengthens the roots of the hair, and, therefore, in cases of alopecia, covers the bald head with hair. As a potion, it helps considerably the extraction of thick and viscous material from the chest and lungs; it breaks up stones and is diuretic. It is helpful in epilepsy and ailments of the spleen. It disperses tumours [4] (p. 497), after [5].

In 1600, Schwenckfeld [6] knew as many as three medicinal moss species identified by us [3] as *Polytrichum commune* Hedw., *P. formosum* Hedw. and *Funaria hygrometrica* Hedw., and 2–4 other nonmedicinal mosses [3], which were to be distinguished from. Ölhafen [7] (p. 164) discovered that *Thuidium tamariscinum* (Hedw.) Schimp. was a hydragoge plant, and this information was absent in the first edition of his book released in 1643 [8].

Moreover, the very beginning of the 17th century saw surprisingly many discoveries of relatively rare and inconspicuous bryophyte species. Let us try to trace the careers of some moss species as candidates for medicinal plants. The following three mosses were described and imaged in 1616 in a botanical work by Colonna [9]:

1. *Targionia hypophylla* L. was discovered by Colonna [9] (p. 333) and named *lichen acaulis hypophyllocarpos*, renamed as *lichen petraeus minimus fructu orobi* [10] (p. 362), and *lichen petraeus minimus acaulis* [11] (p. 1315). All these authors provided only botanical descriptions, and florists of the 18th century, who normally depicted medicinal plants in their floras, did not mention medicinal properties, either. Rupp [12] (p. 294) found *T. hypophylla* near Jena, Goüan [13] (p. 453) near Montpellier, and Withering [14] (p. 701) in the British Isles. It was considered doubtful in Britain [15] (p. 287) (due to its extreme rarity) until it was reported indisputably from Suffolk and Devonshire. *T. hypophylla* is frequent only in southern Europe;

2. *Lichen primus Plinii pileatus* is *Conocephalum conicum* (L.) Underw., according to [16] (p. 184). Colonna [9] (p. 330) stressed: “not mentioned by Dioscorides”. It was used in the same way as *Marchantia polymorpha* L. at least due to its similar habit and habitat [16] (pp. 195–196), and later became an independent article of materia medica named *herba Marchantiae conicae* [17] (p. 188). Medicinal usage of *C. conicum* dates back to the 18th century, as we can judge from this polynomial: *hepatica vulgaris major vel officinarum Italiae*. This name, established by an Italian botanist Micheli [18] (p. 3), emphasizes the presence of *C. conicum* in therapy and drugstores. Raddi [19] (p. 8) proposed the binomial *Fegatella officinalis* Raddi. This taxon was likely mentioned as *Conocephalus officinalis* by Trevisan [20] (p. 785) on his list of Italian liverworts. All these botanical names, as well as the nationality of their authors, reveal the continuous use of this species in Italian medicine;

3. *Lichen alter minor caule calceato hypodedemenos* [9] (p. 332) is *Pellia epiphylla* (L.) Corda (=*Jungermannia epiphylla* L.) [21] (p. [vi]), [22] (p. 7), a plant which is frequent but rather difficult to distinguish from other related thalloid species, and therefore we do not find further mentions of it in European pharmacy.

The examples of *T. hypophylla* and *P. epiphylla* prove that the rarity and small size of the plants (as well as low mass of a potential herbal material) prevented them from being proposed as medicinal plants. Only *C. conicum*, which was discovered in the same year, 1616, became embedded in Italian pharmacy in the early years of the 18th century.

Similarly, common species of mosses could be retained in therapy in external applications. An example is *Brachythecium rutabulum* (Hedw.) Schimp., which was probably recommended in wound dressing at least between 1651 and 1731 [23]. It was a common species; a weed in gardens [24] (p. 444).

In the 18th century, the economical and medicinal value of bryophytes and lichens became a subject of medical dissertations, e.g., by [25,26,27], but no new pharmacological activities were discovered.

## 2. Aim of the Work

This paper reviews the history of the knowledge of medicinal bryophytes between the 16th and 19th centuries as seen in the pharmaceutical, ethnomedicinal and medical works printed in Europe and North America. We present these old records against the background of modern ethnopharmaceutical, pharmacological and phytochemical data to show both continuations in use and knowledge gaps, i.e., species which deserve in-depth research of their pharmacological potential. The ecology of bryophytes (i.e., their abundance, habitats) is also a factor shaping their availability as a potential herbal stock. We gathered data on medicinal bryophytes from historical botanical and pharmaceutical prints (containing pharmaceutical or ethnomedical mentions), as well as from many economical and medical botany reviews (which were frequently published in the 19th century).

Having identified the species mentioned in the original texts, their historical medicinal applications were interpreted and cross-checked with ethnopharmacological monographs (especially [5] and [28]), as well as the most recent data on their chemical components and pharmacology by [28,29,30,31,32].

## 3. Materials and Methods

We queried printed botanical and pharmaceutical publications for mentions of medicinal plants identifiable as mosses or liverworts. They included: botano-medical books (aka *herbals*), dispensatories, pharmacopoeias, formularies, *materia medica* handbooks and pharmaceutical *taxa* (i.e., price-books).

Old polynomials of bryophytes (from pre-Linnaean period in botany) usually contained the Latin nouns *hepatica*, *muscus*, *adiantum* and *polytrichum*. We tracked their subsequent synonymic polynomials in early taxonomical works (by Dillen, Linnaeus and Hedwig), which eventually led us to first binomial basonyms and further binomial nomenclature. Such partial results have been already published by us [3,8,23]. We queried databases (ScienceDirect, Scopus, PubMed) for the identified binomials to find data on their chemical compounds and their pharmacology. Independently, we searched ethnomedicinal and ethnopharmaceutical sources for any mentions of historical or modern use of these species in folk medicine. Types of these sources included: so called natural histories of materia medica, botanical reviews of all known economic plants, reports from scientific expeditions and yearly reports on new ethnomedicinal records (usually titled as new remedies).

“No data” for a given species in Table 1 means that data on chemical composition, pharmacology and uses were absent in the queried databases.

## 4. List of Historical Medicinal Mosses and Bryophytes

Results are presented in Table 1. It contains 7 identified species of liverworts and 21 species of mosses. Taxonomical and ecological commentary on the species is provided below.

### 4.1. Taxonomy and Ecology of Historical Medicinal Bryophytes and Their Position in Pharmaceutical Sciences

#### 4.1.1. Liverworts

*Aneura pinguis* (L.) Dumort. is frequent in calcareous habitats but grows in small populations (forming small turfs) or grows as single plants among other bryophytes [84]. It is similar to other members of the subclass *Metzgeriideae*, which all were scarcely used medicinally (compare the example of *Pellia*, below).

*Conocephalum conicum* (L.) Underw. Gleditsch [85] communicated clearly that the Latin term planta officinalis (‘apothecary plant’) denoted plant materials stored permanently in drugstores, and planta medica (‘medicinal plant’) described any other medicinal species which could be prescribed by a physician. The only bryophyte ever called an apothecary plant was *Conocephalum conicum*, named *Fegatella officinalis* by Raddi [19] in 1818, and *Conocephalus officinalis* by Trevisan [20] in 1874. These binomials, proposed (as nomina nuda) by two botanists of the 19th century (both Italians) prove that *C. conicum* could play a significant role in Italian pharmacy, and that it deserved to be stored in stock in drugstores (Latin: officina). Moreover, the historical internal uses of *C. conicum* are similar to those of *Marchantia polymorpha* L. and these species are similar, too. Their external similarity was noticed by Lightfoot [86] (p. 799) who even compared their pharmacological actions: “the same qualities as *M. polymorpha*, but to a higher degree. Used to treat or prevent scurvy and to thin the blood” [86]. In the late 1880s, it was rated as a powerful diuretic remedy in urolithiasis [87].

*Diplophyllum albicans* (L.) Dumort. An infrequent montane liverwort from Europe [33], forming the most common populations capable of yielding herbal material.

*Marchantia chenopoda* L. does not grow in Europe (Czerwiakowski [39] and Rosenthal [37] reported it mistakenly from a European pharmacy). Its distribution range includes the Caribbean, Middle America, South America, Southern Asia and Oceania, i.e., tropical and subtropical areas of the Americas [88]. Its thalli as a herbal stock were not likely to be imported to Europe.

*Marchantia polymorpha* L. is a liverwort known since antiquity [89]. It is a ubiquitous species, occurring mainly on man-influenced sites worldwide, except for the Arctic [84]. Today, it is the most widely used liverwort and one of the most cited bryophytes to have medicinal potential [5]. Taxonomy of this polymorphic species is tangled. Bischler-Causse and Boisselier-Dubayle [90] divided *M. polymorpha* into three subspecies: *M. p*. subsp. *polymorpha* (Nees) Burgeff, *M. p*. subsp. *montivagans* Bischl. & Boissel.-Dub. and *M. p.* subsp. *ruderalis* Bischl. & Boissel.-Dub. Other authors (e.g., Damsholt [84]) consider them as three closely allied species: *M. aquatica* (Nees) Burgeff (=*M. p.* subsp. *polymorpha*), *M. alpestris* (Nees) Burgeff (=*M. p.* subsp. *montivagans*) and *M. latifolia* Gray (=*M. p.* subsp. *ruderalis*). Nowadays, most taxonomists agree that *M. polymorpha* is a single species with three subspecies (e.g., [91,92]): *M. p.* subsp. *polymorpha* grows infrequently in wet places (banks of streams, springs, peat bogs), *M. p.* subsp. *montivagans* in wet places in mountains and *M. p.* subsp. *ruderalis* is the commonest subspecies of human-altered ecosystems, ubiquitous and usually reported jus as “*M. polymorpha*”. It is interesting that in phytochemical research none of these subspecies were ever mentioned; hence, we do not know any potential chemical differences among these subtaxa. Regarding the historical ethnopharmacology, *M. polymorpha* tastes bitter [56] (p. 367). As the taste resembles bile, it could determine folk use in liver conditions (according to the doctrine of signatures).

*Pellia epiphylla* (L.) Corda is a similar case. Ethnomedical usage of liverworts from this genus (no species given) is known from Canada. Among Hesquait Indians, “this liverwort was used if a child had a sore mouth or throat that prevented them from eating or drinking. They would either drink the juice or chew the pulp of the liverwort” [93] (p. 135). Nitidaht (Ditidaht) Indians used it internally and/or externally “for pain anywhere in the body” [45] (p. 135).

*Reboulia hemisphaerica* (L.) Raddi—a liverwort mostly from southern Europe, regarded as rare [84]. Its populations are mainly small, so it cannot afford much herbal material.

*Targionia hypohylla* L., described botanically by Colonna [9] in 1616, lacks a testimony from European ethnopharmacology. However, it was recorded in traditional Hindu medicine for curing scabies, itches and other skin diseases by the Irula tribes [94]. Its strong antifungal and antibacterial properties were recently found [95] and thus its efficacy in wound healing was scientifically confirmed [96].

#### 4.1.2. Mosses

*Brachythecium rutabulum* (Hedw.) Schimp. is a large moss, growing in lax, glossy, bright green or yellowish green tufts or patches. It is common in Europe and occurs in many habitats, such as soil (both in woodland and non-forest vegetation), tree boles, logs, stones and walls [97]. It is frequently found in man-made habitats such as lawns or gardens, where it is regarded as an unwanted plant. *B. rutabulum* deserves a pharmacological study, because [98] recently reported antimicrobial activity of an allied species, *B. campestre* (Müll. Hal.) Schimp.

*Fontinalis antipyretica* Hedw., one of Europe’s largest mosses, grows sometimes in large clusters in clean streams and lakes [97], from where it can be harvested with ease.

*Funaria hygrometrica* Hedw., a tiny moss species, but sometimes very abundant in man-made habitats. It seems easy to gather and was one of the first mosses mentioned in floras and in medicinal and botanical works (e.g., [6,99]; see [3]).

*Grimmia pulvinata* (Hedw.) Sm. (=*Dryptodon pulvinatus* (Hedw.) Brid.) grows on stones, walls and concrete constructions. Easily available, one of the commonest European members of the *Grimmiaceae* family [97].

*Homalothecium sericeum* (Hedw.) Schimp. is moderately robust, glossy, yellowish green to golden brown and occurs in dense rough mats or patches, mainly on the bark of trees and on bare rocks. Sometimes it grows on man-made habitats such as walls and roofs, and it is common in Europe [97].

*Meesia*: members of this genus grow in peat bogs, mainly in northern Europe, but are rare outside this area, being glacial relics. They seem therefore inaccessible for harvesting, and these species might be misidentified.

*Polytrichum commune* Hedw., a conspicuous moss, the largest representative of its genus, frequent in Europe in moist habitats [97]. Used in ethnopharmacies worldwide [5].

*Polytrichum formosum* Hedw. and ***P. longisetum*** Brid. are frequent in forests of Europe [97] and are of large size, and therefore seem to be easily harvested (especially *P. formosum*).

*Polytrichum juniperinum* Hedw. is another conspicuous species of this genus, which prefers dry and acidic habitats, frequent in Europe [97].

*Ptilium crista-castrensis* (Hedw.) De Not.—a forest species, sometimes abundant in European coniferous forests [97].

*Hylocomiadelphus triquetrus* (Hedw.) Ochyra & Stebel, *Rhytidiadelphus loreus* (Hedw.) Warnst., *Rh. squarrosus* (Hedw.) Warnst., *Hypnum cupressiforme* Hedw. and *Pleurozium schreberi* (Brid.) Mitt. are large moss species, growing mostly in forests, thickets and meadows, all common across Europe [97]. They seem easy to harvest as herbal stock.

## 5. Discussion

Historical ethnobiological and ethnopharmaceutical uses of bryophytes are sometimes convergent with our contemporary knowledge, e.g., in the case of many bryophytes used externally which prove to be antimicrobial [28,29,31]. However, the greatest number of biochemical and pharmacological reports on European bryophytes concern *Marchantia polymorpha* [28,30,31,46].

The persistence of *Conocephalum conicum* in the official Italian pharmacy of the 19th century is peculiar. Therefore, *C. conicum* deserves pharmacological research as deep as that of *M. polymorpha*.

The only description of *Marchantia chenopoda* as a remedy was published by [55] from Cuba in 1889, probably based on earlier observations or testimonies. Its usage in liver affections by [37,38,55] is convergent or may originate from an unknown earlier source. Diuretic action in hydrops was cited by de la Maza [55] according to a physician named Short.

The following bryophyte species mentioned in this article form small populations which make their harvesting rather difficult and improbable: *Aneura pinguis*, *Meesia* spp., *Reboullia hemispherica*, *Diplophyllum albicans* and *Targionia hypophylla*. Abundance of the following species makes their harvesting possible and effective (and has done so throughout the centuries): *Brachythecium rutabulum*, *Conocephalum conicum*, *Fontinalis antipyretica*, *Funaria hygrometrica*, *Grimmia pulvinata*, *Homalothecium sericeum*, *Marchantia polymorpha*, *Polytrichum* spp., *Ptillium crista-castrensis*, *Rhytidiadelphus* spp., *Sphagnum* spp. and *Thuidium tamariscinum*.

We are still lacking any biochemical and pharmacological data on the following European bryophyte species used here in the past centuries: *Aneura pinguis* (L.) Dumort., *Brachythecium rutabulum* (Hedw.) Schimp. and *Meesia uliginosa* Hedw.

Our knowledge of many further species seems incomplete when related to many ethnopharmacological reports on their usage, and it requires assessment. Here belong: *Pleurozium schreberi*, *Rhytidiadelphus loreus*, and *Thuidium tamariscinum*.

## 6. Conclusions

1. Historical European works on vegetable materia medica and therapy from the years 1616–1889 mention 28 identifiable species of bryophytes (21 mosses, 7 liverworts and no species of hornworts). Only three of them (*Aneura pinguis*, *Brachythecium rutabulum* and *Meesia uliginosa*) have not been searched for pharmacologically active substances so far;

2. *Conocephalum conicum* was the only bryophyte species of wider therapeutical usage in the 18th and 19th centuries (especially in Italian pharmacy);

3. *Marchantia chenopoda* was reported as a medicinal plant from Cuba in 1889, but prior to this year it was mentioned twice by European materia medica writers (1849 and 1862) as a diuretic and in liver aliments, probably based on earlier clinical or ethnopharmaceutical testimonies. Its herb was never seen in Europe;

4. *Anthocerotophyta* is the division of bryophytes which gathers minute organisms growing in small populations. They have never been used medicinally [5] (p. 186). However, ethnopharmaceutical information was recently published about the use of *Ceratophyllum demersum* L. as a purgative, astringent, constipating and antipyretic [28,45]. *C. demersum* was erroneously classified among hornworts (i.e., bryophytes) due to its English name *rigid hornwort*. It is a flowering plant of the *Ceratophyllaceae* family which contains water plants growing in tropical and temperate regions [100]. Hence, *Anthocerotophyta* remain the only division of bryophytes without known ethnopharmacological uses.

## Figures and Tables

**Table 1 plants-10-01296-t001:** Historical medicinal bryophytes, their taxonomical identification, ethnopharmaceutical uses and confirmed medicinal properties.

Original (Pharmaceutical) Names of Plants or Herbal Material	Original Applications	Sources for Original Applications	Species Identification (Current Name According to [33])	Uses in Traditional Medicine	Contemporary Knowledge
*Jungermannia pinguis* L.	It contains iodine and was used to obtain it.	[34] (p. 38)	*Aneura pinguis* (L.) Dumort.	No data	No data
*muscus terrestris et hortensis*	Stemming the blood	[35] (p. 764), [24] (p. 444), [27] (p. 15)	*Brachythecium rutabulum* (Hedw.) Schimp. [23]	In the 17th–18th centuries, used for dressing wounds [23]	No data
*Marchantia conica, Hepatica stellata, H. fontana.*	Aperitive, acrid, astringent, used in diseases of the liver.	[36] (p. 11)	*Conocephalum conicum* (L.) Dumort. (including newly described *C. salebrosum* Szweyk., Buczk. and Odrzyk.)	Antimicrobial, antifungal, antipyretic, antidotal activity; used to cure cuts, burns, scalds, fractures, swollen tissue, poisonous snake bites and gallstones [28,29]	Antimicrobial, antifungal and antiviral activity, superoxide release inhibitory Sactivity [31]. Antimicrobial [37]. Insecticidal [38]
*Herba Marchantiae conicae, music hepatici*	Thick decoct of the herb. Cataplasms on swollen parts. Analgesic. Diuretic in calculus and hydrops.	[17] (p. 188)			
*Fegatella conica*	Resolving and purging organs, especially the liver	[39] (p. 160)			
*herba Hepaticae fontinalis = herba Lichenis stellati Fegatella officinalis* Raddi = *Marchantia conica* L.	Resolving, and in conditions of the liver	[34] (p. 39)			
*Fontinalis antipyretica*	The herb boiled with beer applied as a footbath (*pediluvium*) in pectoral fever	[40] (p. 379), [26] (p. xxv)	*Fontinalis**antipyretica* Hedw.	Chest fever, antimicrobial, to treat fever and for detoxification [41,42,43]	Antimicrobial and antiproliferative [44]. Anti-proliferation activity against neoplastic cell lines: rat glioma (C6), and antifungal [45] Antimicrobial and antifungal [46]
*Fontinalis antipyretica*	In fevers	[47] (p. 9)			
*Fontinalis antipyretica*	For *pediluvia* (i.e., foot wash)	[27] (p. 12), [48] (p. 33)			
*herba Fontinalis antipyreticae*, *Fontinalis antipyretica* L.	In breast ailments, decoct in footbaths	[34] (p. 39)			
*Fontinalis antipyretica*	Decoct was once recommended in pulmonary conditions and as a footbath	[39] (p. 183)			
*Funaria hygrometrica*	Diaphoretic, diuretic, laxative	[39] (p. 172)	*Funaria hygrometrica* Hedw.	Hemostatic, in pulmonary tuberculosis, *vomitus cruentus* (hematemesis), bruises and athlete’s foot dermatophytosis (dermatomycosis) [29,41]	Antiproliferative to cancer cells: HeLa (of cervical cancer), A2780 (ovary cancer), T47D (breast cancer) [49]. Slightly antibacterial against *B. subtilis*, *P. aeruginosa* and *S. aureus*; antifungal [50] and antibacterial [51].
*Bryum pulvinatum* L.; *Dryptodon pulvinatus*	Immersed in vinegar and imposed onto the top of the head and into the nostrils, stops hemorrhage; poultices of the herb boiled with vinegar	[27] (p. 14), [48] (p. 33)	*Grimmia pulvinata* (Hedw.) Sm.	No data	High activity against 3 Gram-positive bacteria: *E. faecalis, S. aureus* and *S. pyogenes*, and 2 Gram-negative bacteria: *E. coli* and *K. pneumoniae* [47]
*Hypnum sericeum. Leskia sericea*, *Leskea sericea* Hedw.	In hemorrhages (hemostatic, styptic)	[27] (p. 15), [36] (p. 11), [48] (p. 33), [39] (p. 184), [34] (p. 39)	*Homalothecium sericeum* (Hedw.) Schimp.	No data	Antomicrobial [43,52]. Antiproliferative [53]. Antioxidant [54].
*Jungermannia alba* L.	It contains iodine and was applied to obtain it	[34] (p. 38)	A confused name (non-existent binomial). Maybe *Diplophyllum albicans* (L.) Dumort.	No data	*D. albicans*: antimicrobial, antifungal and antiviral [31]. *P. albescens*: no data.
*Marchantia chenopoda*	In the West Indies, used in liver aliments	[39] (p. 162)	*Marchantia chenopoda* L. This species is not native to the flora of Europe; the herbal raw material was not likely to be imported here	No data	Antimicrobial, antifungal and antiviral [31].
*Marchantia chenopodea* L.	In the (West) Indies: clumps of the abdomen, esp. the liver	[34] (p. 39)			
*Marchantia chenopoda* L.	Tempering, aperitive. In hepatical, renal, vesico–urinary and dermal affections. Externally in hydrops. It acts as a diuretic.	[55] (pp. 3–4)			
*Marchantia polymorpha*, *Marchantia polymorpha* L.	In ailments of liver, in jaundice, to promote lactation	[26] (p. xvii), [47] (p. 109)	*Marchantia polymorpha* L.	Antipyretic, antihepatic, antidotal, diuretic activity; used to cure cuts, fractures, poisonous snake bites, burns, scalds and open wounds [28,29]	Cytotoxic, antimicrobial, antifungal and antiviral, inhibitory activity against 5-lipoxygenase, calmodulin, hyaluronidase, cyclooxygenase, DNA, polymerase b and a-glucosidase, cathepsin L and cathepsin B inhibitory activity [31]. Antifungal activity [45].
*hepatica terrestris, Marchantia polymorpha* L.	Aperient, resolvent, antiscorbutic	[56] (p. 367)			
*Marchantia polymorpha* L.	A resolving remedy in ailments of the liver	[34] (p. 39)			
*Meesia uliginosa*	Promoting sweat, urine and stool	[39] (p. 178)	*Meesia uliginosa* Hedw.	No data	No data
*Adiantum aureum Tabern., percemousse, golden Wiederthon.*	Roborant	[57] (p. 372)	*Polytrichum commune* Hedw.	Antipyretic and antidotal; for hemostasis, cuts, bleeding from gingivae, hematemesis and pulmonary tuberculosis [29]. *P. commune* is a traditional Chinese remedy for fever, hemorrhage, uterine prolapse and lymphocytic leukemia [44].	Cytotoxic [58]. Anti-cancer and proapoptotic [44]. Antibacterial [59]. Anti-neuroinflammatory [60].
*Polytrichum commune α*	A decoct nourishes and strengthens the hair, applied externally	[27] (p. 13)			
*Polytrichum commune*	It belongs to the five capillary herbs	[61] (p. 7)			
*Polytrichum commune*	In cough and pleurisy, in form of syrup, decoct or distilled water and then sudorific. Boiled in red wine as emmenagogue. For liver obstruction and jaundice.	[26] (pp. xxvi–xxvii)			
*Polytrichum commune* L.	Antitussive, sudorific, antipleuritic; drug form: distilled water.	[62] (p. 6)			
*Polytrichum commune*	In pleuritis	[63] (No. 343)			
*Polytrichum commune*	In jaundice, urolithiasis, colic.	[47] (p. 218)			
*herba adianthi aurei* constituted by *Polytrichum commune* L.	An astringent, in diseases of the lungs, and calculous complaints	[56] (p. 18)			
*adiantum aurerum* = *Polytrichum vulgare*	Sudorific, pulmonary	[36] (p. 11)			
*Adiantum aureum seu majus = Polytrichum commune*	Diaphoretic	[64] (p. 52)			
*Polytrichum commune*	Decoct or ethereous tincture as emmenagogue and lactagoge.	[17] (pp. 231–232)			
*herba Polytrichi = hb. musci capillaris*	Gently astringent, promoting sweat and urine	[39] (p. 180)			
*herba adianta aurei, Polytrichum commune*	Mucilaginous	[65] (p. 155)			
*Polytrichum commune*	Veterinary	[66] (p. 83)			
*Polytrichum juniperum*	Powerful diuretic	[67] (pp. 323–324)	*Polytrichum juniperinum* Hedw.	Prostate and urinary difficulties; skin ailments [28]	Antibacterial action against *Bacillus cereus, Enterococcus faecalis*, *Staphylococcus aureus* and *Streptococcus pyogenes* [68]. Cytotoxic and antiproliferative against the following lines of neoplastic cells: rat glioma (C6), human epidermoid carcinoma (A431), human lung cancer (A549), human breast cancer (MCF-7), human intestinal cancer (CaCo-2) and mouse Sarcoma 37 cells. [69].
*Hypnum castrense*	Chewed and applied onto snake bites.	[63] (No. 343)	*Ptilium crista-castrensis* (Hedw.) De Not.	No data	Antibacterial against *Proteus mirabilis*, *Bacillus cereus*, *Escherichia coli*; antiproliferative against lines of neoplastic cells: rat glioma (C6), human epidermoid carcinoma (A431), human lung cancer (A549), *Mus musculus* skin melanoma (B16-F10), human breast cancer (MCF-7) and human intestinal cancer (CaCo-2) [69].
*Rebouillia hemisphaerica* *vel androgyna*	In England, it is used in edema (hydrops)	[39] (pp. 159–160)	*Reboulia hemisphaerica* (L.) Raddi	For blotches, hemostasis, external wounds and bruises [29]	Liver X-receptor (LXR)a agonist and (LXR)b antagonist activity [30]. Antiplatelet activity [70]. Antimicrobial [71].
*Marchantia hemisphaerica* L.	Conditions of liver, and hydrops	[34] (p. 1070)			
*muscus vulgaris* in which were recognized: *Hypnum triquetrum* L., *H. loreum* L., *H. schreberi* Willd., *H. sqarrosum* L., and *H. cupressiforme* L.	Moderately astringent, subdiaphoretic, gently diuretic	[48] (p. 33)	1. *Hylocomiadelphus triquetrus* (Hedw.) Ochyra & Stebel2 2. *Rhytidiadelphus loreus* (Hedw.) Warnst. 3. *Pleurozium schreberi* (Brid.) Mitt. 4. *Rhytidiadelphus squarrosus* (Hedw.) Warnst. 5. *Hypnum cupressiforme* Hedw.	No data	*H. triquetrus*: antiproliferative and cytotoxic [72]. Antimicrobial [73]. *H. cupressiforme*: antimicrobial, antifungal [47] and insecticidal [39].
*Herba adianti aurei = hb. Polytrichi = hb. musci capillacei*, which was constituted by: *Polytrichum commune* L., *P. formosum* (Hedw.) G. L. Sm., *P. longisetum* Sw., *P. juniperifolium* Hoffm., *Funaria hygrometrica* Hedw., *Meesia uliginosa* Hedw.	Moderately astringent, subdiaphoretic, gently diuretic	[48] (p. 33)	1. *Polytrichum commune* Hedw. 2. *Polytrichastrum formosum* (Hedw.) G. L. Sm. 3. *Polytrichastrum longisetum* (Brid.) G. L. Sm. 4. *Polytrichum juniperinum* Hedw. 5. *Funaria hygrometrica* Hedw. 6. *Meesia uliginosa* Hedw.	Various species of *Polytrichum*: diuretic; promoting the growth of hair [29]	For *F. hygrometrica*, *M. uliginosa*, *P. commune*, *P. juniperinum* see above. *P. formosum* is antimicrobial [74].
*Sphagnum palustre* L.	As a diaper and bedding for nursing babies, a good urine absorbent	[25] (p. 9), [75] (p. 1445), [76] (p. 100)	*Sphagnum palustre* L., later *Sphagnum* spp.	No data	*Sphagnum* spp.: odor absorbent for personal hygienic uses, and a historical but powerful wound dressing material [77]. A polysaccharide compound called sphagnan induces Maillard reaction with proteins, lowers the pH and binds nitrogen compounds (including amino-acids and enzymes) [78,79]. Food preservation [80].
*Sphagnum palustre* L.	Humid and frigid, applied to swollen feet in elderly people	[40] (p. 379)			
*Sphagnum palustre* L.	In foot edema: as a decoct made from the herb and beer, applied as a poultice	[26] (p. xxv), [27] (p. 12)			
*Sphagnum* spp.	Dressing material	[81] (p. 12)			
*Sphagnum fimbriatum* Wils., *S. squarrosum* Pers., *S. strictum* Lindb.	Baby bedding, diapers	[82] (p. 204)	*Sphagnum fimbriatum* Wilson, *S. squarrosum* Crome, *S. girgensohnii* Russow, respectively.		
*muscus filicinus major*	Hydragoge (diuretic)	[7] (p. 164)	Thuidium tamariscinum (Hedw.) Schimp. [8]	No data	Antioxidant [83].

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
