# Peer review of "Four Centuries of Medicinal Mosses and Liverworts in European Ethnopharmacy and Scientific Pharmacy: A Review"

_plants, 2021, doi:10.3390/plants10071296_

Round 1

Reviewer 1 Report

  1. I have highlighted lines where the authors need to cite references properly. File uploaded.
  2. The conclusions are more or less repeating what is already in the text - it is not necessary and could be deleted. The editor should decide that.

Author Response

REVIEWER NO. 1 WROTE:

I have highlighted lines where the authors need to cite references properly. File uploaded.

AUTHORS' ANSWER:

Thank you very much for pointing out the citation problems. We have inserted names of authors in the highlighted places, so that each sentence has got a grammatical subject.

REVIEWER NO. 1 WROTE:

The conclusions are more or less repeating what is already in the text - it is not necessary and could be deleted. The editor should decide that.

AUTHORS' ANSWER:

We have deleted the Conclusion No. 1. as concerning earlier times than studied in this manuscript. I'd keep the other conclusions as they are hard to draw from the rough text of Table 1, and the last one is our correction of the data already published by someone.

Reviewer 2 Report

Authors should better describe the "matherial and methods" used, with  deeper description of the original texts where plants were selected, the methodology and tools applied for plant identification, and the reference for taxonomical identification.

A comment on no data of contemporary knowledge is encouraged. Do authors intend that no data are available on references (5, 28-31) or no data are available on classical scientific database (I.e. pubmed, scopus, etc.)

A significative improvement of the contemporary knowledge should also derives from the inclusion of relevant information such as type of extract(s) and chemical charcterization (identification of main componds or at least in terms of presence/absence of chemical fingerprint) 

Author Response

REVIEWER NO. 2 WROTE:

Authors should better describe the "matherial and methods" used, with  deeper description of the original texts where plants were selected, the methodology and tools applied for plant identification, and the reference for taxonomical identification.

AUTHORS' ANSWER

Thank you for this remark. We have now added the Materials and methods section presenting types of sources used, as well as the pre-taxonomical and taxonomical methods of plant identification used by us.

REVIEWER NO. 2 WROTE:

A comment on no data of contemporary knowledge is encouraged. Do authors intend that no data are available on references (5, 28-31) or no data are available on classical scientific database (I.e. pubmed, scopus, etc.)

AUTHORS' ANSWER

In the Materials and methods part, we added a remark that the "no data" statement in Table 1 means lack of pharmacological and ethnopharmacological information in modern scientific databases.

REVIEWER NO. 2 WROTE:

A significative improvement of the contemporary knowledge should also derives from the inclusion of relevant information such as type of extract(s) and chemical charcterization (identification of main componds or at least in terms of presence/absence of chemical fingerprint) 

AUTHORS' ANSWER

The "relevant information such as type of extract(s) and chemical characterization (identification of main compounds or at least in terms of presence/absence of chemical fingerprint)", if any is available, would make our text very lengthy, instead we referenced experimental and review articles in the last column. For example, the paper [84] contains as many as 53 compounds found in extracts of only two species (P. commune and Rh. triquetrus). In our opinion, such a detailed chemical approach would be more applicable in reviews devoted to a single species or genus or to a particular type of pharmacological action. Instead, we focused on the historical and taxonomical aspect of finding, identification and the importance of a bryophyte species in therapy and folk medicines worldwide. Misidentifications of bryophyte species occur today (see our last conclusion) and are almost unfeasible for botanists who never studied historical taxonomy, historical therapy and historical pharmacy.